# Privacy-Aware Time Series Synthesis via Public Knowledge Distillation

**Penghang Liu**                                    *penghang.liu@jpmchase.com*
*JPMorgan AI Research*

**Haibei Zhu**                                      *haibei.zhu@jpmchase.com*
*JPMorgan AI Research*

**Eleonora Kreacic**                                *eleonora.kreacic@jpmchase.com*
*JPMorgan AI Research*

**Svitlana Vyetrenko**                              *svitlana.s.vyetrenko@jpmchase.com*
*JPMorgan AI Research*

**Reviewed on OpenReview:** *https://openreview.net/forum?id=TC6ihoRwOc*

## Abstract

Sharing sensitive time series data in domains such as finance, healthcare, and energy consumption, such as patient records or investment accounts, is often restricted due to privacy concerns. Privacy-aware synthetic time series generation addresses this challenge by enforcing noise during training, inherently introducing a trade-off between privacy and utility. In many cases, sensitive sequences is correlated with publicly available, non-sensitive contextual metadata (e.g., household electricity consumption may be influenced by weather conditions and electricity prices). However, existing privacy-aware data generation methods often overlook this opportunity, resulting in suboptimal privacy-utility trade-offs. In this paper, we present Pub2Priv, a novel framework for generating private time series data by leveraging heterogeneous public knowledge. Our model employs a self-attention mechanism to encode public data into temporal and feature embeddings, which serve as conditional inputs for a diffusion model to generate synthetic private sequences. Additionally, we introduce a practical metric to assess privacy by evaluating the identifiability of the synthetic data. Experimental results show that Pub2Priv consistently outperforms state-of-the-art benchmarks in improving the privacy-utility trade-off across finance, energy, and commodity trading domains.

## 1 Introduction

Synthetic data generation has emerged as a powerful approach for mitigating the risks associated with sharing sensitive real-world data, leading to the development of numerous learning-based models designed for this purpose (Figueira & Vaz, 2022; Lu et al., 2023; Sezer et al., 2020; Potluru et al., 2023). A de facto standard for ensuring data privacy during training is differentially private stochastic gradient descent (DP-SGD) (Abadi et al., 2016), which provides strong privacy guarantees for individual data points by introducing gradient noise and normalization. Accordingly, various efforts have been made to integrate differential privacy (DP) into generative models (Xie et al., 2018; Papernot et al., 2018). These approaches inherently involve a privacy-utility trade-off, where stronger privacy protection often reduces the usefulness of the generated data for downstream tasks.

Researchers have explored methods to alleviate this trade-off by leveraging additional information that does not compromise privacy. Semi-private learning, for example, integrates publicly available and non-sensitive

auxiliary datasets to enhance the privacy-utility trade-off (Pinto et al., 2024). Previous work such as Wang & Zhou (2020) utilized small amounts of public information to adjust parameters in DP-SGD and fine-tune results using model reuse. Additionally, studies by Alon et al. (2019); Lowy et al. (2024) demonstrate that differential private learning can significantly benefit from auxiliary public data, even unlabeled. Although these methods enhance the model's generative capabilities, they are mainly effective under the assumption that public and private data originate from the same source and share similar distributions. However, this assumption rarely holds in real-world synthetic time series generation scenarios since homogeneous public data are often limited in availability and contain fewer samples than their private counterparts (Jordon et al., 2018).

A critical but often overlooked aspect in privacy-preserving data generation is the heterogeneous contextual metadata (Narasimhan et al., 2024), which can be closely connected to sensitive time series data of interest but does not raise any privacy concerns. For instance, local weather and energy pricing data can serve as valuable public signals for modeling household electricity consumption time series while maintaining the anonymity of individual usage patterns and household-specific information. Similarly, while investment portfolios and personal trading activities are confidential, they exhibit correlations with publicly available stock market indices such as the S&P 500 and Nasdaq Composite. Furthermore, recent studies have highlighted the potential capability of using empirical knowledge embedded in large language models (LLMs) to identify auxiliary datasets (Zhu et al., 2024), thereby exploring and collecting public knowledge that can enhance the privacy-utility trade-off for synthetic data generation. Motivated by this opportunity, we explore a new perspective on privacy-aware data generation by leveraging publicly available contextual knowledge. Through our novel problem formulation, we seek to examine how public knowledge can enhance the privacy-utility trade-off across diverse real-world scenarios and domains.

To the best of our knowledge, no prior work has explored the use of public contextual information for time series generation with differential privacy (DP) guarantees. Moreover, it remains unclear how such information can be leveraged to enhance time series generation performance without incurring additional privacy loss. To address this gap, we introduce Pub2Priv, a diffusion model designed to generate sensitive time series using public knowledge. Our approach utilizes a pre-trained transformer to extract embeddings from multi-dimensional public metadata, which are then used as conditional inputs for a diffusion model to generate synthetic private data. To ensure privacy protection, we train our model using DP-SGD under a specific privacy budget. We also propose a practical metric to evaluate the privacy of our framework by measuring the identifiability of synthetic data. Through experiments across multiple domains, including finance, energy, and commodity trading, we comprehensively evaluate the privacy and utility of the synthetic time series generated by Pub2Priv. Our key contributions are summarized as follows.

- We introduce a novel problem formulation of privacy-aware time series generation by considering publicly available heterogeneous metadata.
- We develop Pub2Priv, a conditional diffusion model framework to utilize public domain knowledge with no additional privacy cost. Our model utilizes self-attention layers to capture temporal and feature correlations in multidimensional contextual metadata, enabling realistic private time series synthesis.
- We introduce a new practical privacy evaluation metric based on the identifiability of the synthetic data, which serves as a more interpretable tool to assess and configure privacy guarantees in generation models.

## 2 Related Works

### 2.1 Synthetic Data Generation Models

Synthetic data generation has been an active area of research, with a broad range of models proposed to capture and reproduce the statistical characteristics of real-world datasets (Dankar & Ibrahim, 2021; Raghunathan, 2021; Assefa et al., 2020). Early efforts focused on Generative Adversarial Networks (GANs) (Goodfellow et al., 2020) for image data. This framework has been extended to various data modalities,

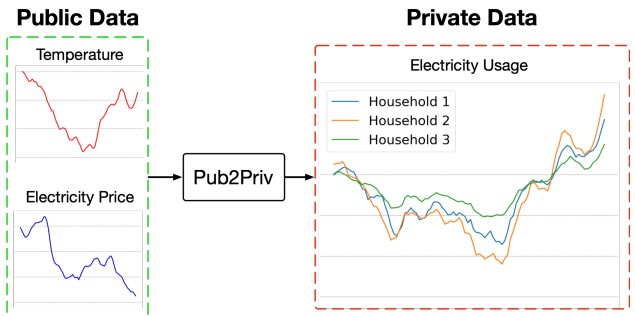

Figure 1: Pub2Priv generates private time series from heterogeneous public knowledge. The model generates realistic electricity consumption based on non-secret temperature and electricity price information. Household private data is protected by DP-SGD during training.

including tabular, text, and time series (Zhang et al., 2017; Xu et al., 2019; Li et al., 2022). For tabular data, methods such as CTGAN use conditional generators to capture complex interactions among categorical and continuous variables. Language models based on transformers have been widely used in text generation and successfully generated high-fidelity sentences or even entire documents (Achiam et al., 2023). Similarly, diffusion models have recently emerged as powerful generative models in various tasks via the interactive denoising process (Ho et al., 2020).

A significant body of work also focuses on time series data generation. For example, Esteban et al. (2017) proposed RCGAN to produce realistic real-valued multi-dimensional time series based on recurrent neural networks (RNNs) and generative adversarial networks (GANs). Yoon et al. (2019) presented TimeGAN, which combines the strengths of unsupervised GANs and supervised autoregressive models for controlling temporal dynamics. Desai et al. (2021) further contributed to this field by introducing TimeVAE, a Variational Auto-Encoder (VAE), to generate multivariate time series with good interpretability. Alaa et al. (2021) presented a novel approach to time series generation by focusing on the frequency domain instead of the time domain. In parallel, recent research in autoregressive modeling leveraged large language models and structured narratives for improved sequence modeling and forecasting (Liu et al., 2024b;a); while primarily targeted at forecasting rather than unconditional generation, these approaches are closely related to learning temporal dependencies.

More recently, diffusion models (Ho et al., 2020; Song et al., 2020) have emerged as a promising approach for time series generation. Tashiro et al. (2021) introduced the CSDI model for time series imputation, showcasing the potential of diffusion models in handling missing data within temporal sequences. Building on this, Narasimhan et al. (2024) developed TIME WEAVER, a model specifically tailored for time series generation that leverages metadata to enhance the generative process. These diffusion-based approaches highlight the evolving landscape of time series generation, where models increasingly incorporate context and external information to generate realistic and useful synthetic data.

## 2.2 Privacy-Aware Learning

Privacy-aware learning has gained significant attention in recent years, particularly in protecting sensitive information while training machine learning models. Abadi et al. (2016) introduced Differentially Private Stochastic Gradient Descent (DP-SGD), a method that has since become a standard for training models with privacy guarantees. DP-SGD works by applying gradient clipping and adding noise to the gradients during training, ensuring that the privacy of individual data points is preserved. Similarly, Yu et al. (2021) proposed gradient embedding perturbation techniques, which project gradients into a lower-dimensional space before applying noise, reducing the dimensionality of perturbations and improving model utility while maintaining privacy.

Semi-private learning has emerged as an effective approach to leverage both public and private data for training models with enhanced accuracy (Pinto et al., 2024). Alon et al. (2019); Wang & Zhou (2020)

explored the limits and potentials of semi-private learning, demonstrating that public data can be effectively used to adjust model parameters and improve the outcomes of differentially private learning. Lowy et al. (2024) further extended this by proposing optimal strategies for model training with public data, while Amid et al. (2022) discussed the benefits of using in-distribution versus out-of-distribution public data in privacy-preserving machine learning. Another notable approach is the Private Aggregation of Teacher Ensembles (PATE) framework proposed by Papernot et al. (2016; 2018), where teacher models trained on sensitive data transfer knowledge to a student model trained on public data, effectively balancing privacy and utility. While semi-private learning approaches provide an opportunity to utilize public data in private data generation, most existing studies treat public information as an additional dataset rather than as heterogeneous contextual signals. In time series specifically, work under differential privacy has centered on forecasting (Arcolezi et al., 2022; Schuchardt et al., 2025) or on non-DP signal-processing style anonymization via multivariate all-pass filtering (Hore et al., 2024), whereas the DP time-series generators are usually unconditional DP-GAN without external context (Frigerio et al., 2019). This leaves the use of heterogeneous public context for generation under DP largely unexplored.

Despite these advancements, the application of privacy-aware learning techniques to synthetic data generation, particularly in complex domains such as images and time series, remains a challenging task. Early attempts, such as Xie et al. (2018), incorporated DP-SGD into the GAN framework, while Jordon et al. (2018) developed a GAN model based on the PATE framework. For time-series data specifically, Frigerio et al. (2019) proposed a DP-GAN with an RNN-based generator to release synthetic sequences, but without conditioning on public metadata. More recently, Dockhorn et al. (2022) extended this line of research by applying DP-SGD to diffusion models, showcasing the adaptability of differential privacy across different generative paradigms.

An alternative approach to privacy-aware data generation is ADS-GAN, proposed by Yoon et al. (2020), which aims to generate synthetic data while minimizing patient identifiability. However, this method is more aligned with data anonymization rather than true data generation, as it requires real data as input to produce synthetic outputs. The focus of our work is on developing methods for data generation rather than data anonymization or obfuscation. Specifically, we aim to address the gap in existing research by introducing a model that generates synthetic time series data using heterogeneous public data, offering a novel approach to privacy-preserving data generation that leverages external information.

In parallel, another line of research focuses on generating synthetic data for query release by preserving marginal statistics. Notable examples include GEM (Liu et al., 2021), which optimizes generative neural networks using the exponential mechanism, AIM (McKenna et al., 2022), which iteratively selects queries in a workload-adaptive manner, and Private-GSD (Liu et al., 2023), a zeroth-order optimization approach based on genetic algorithms. These methods aim to ensure high-fidelity answers to statistical queries under differential privacy constraints, offering strong empirical performance in structured data domains.

## 3 Preliminaries

### 3.1 Differential Privacy

In this study, we aim to protect the privacy of individual data $x \in D_x$, such that given the data generated by our model $D_{x'} = G(D_X)$, an attacker can not identify any data point in the original data (i.e., can not certainly tell whether $x \in D_x$). One way to provide a strong guarantee of individual privacy is Differential Privacy (DP), which is defined as follows.

**Definition 3.1** ($\varepsilon, \delta-$Differential Privacy). (Dwork et al., 2006) A randomized mechanism $\mathcal{M} : D \to \mathcal{R}$ with domain $\mathcal{D}$ and range $\mathcal{R}$ is $(\varepsilon, \delta)-$ differentially private if for any two neighboring datasets $D, D' \in \mathcal{D}$ that differ by at most one element, and for any subset of output $S \subseteq \mathcal{R}$, it holds that,

$$\Pr[\mathcal{M}(D) \in S] \leq e^{\varepsilon} \Pr[\mathcal{M}(D') \in S] + \delta$$

DP offers strong privacy guarantees by ensuring that the inclusion or exclusion of a single data point does not significantly affect the model's output. In the equation, $\varepsilon$ is the privacy budget and $\delta$ denotes the

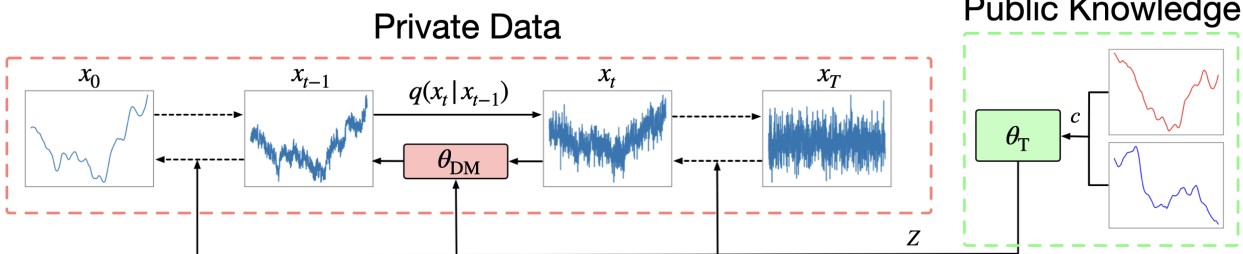

Figure 2: Pub2Priv architecture. Given the original data sample $x_0$, we gradually add noise through forward process $q(x_t|x_{t-1})$. In the reverse process, we use self-attention layers $\theta_T$ to create temporal and feature embedding of the metadata $c$, which is passed to the conditional denoiser $\theta_{DM}$ to reconstruct the original sample.

probability of a privacy breach (in practice, $\delta$ is typically set to be smaller than $1/|\mathcal{D}|$)). The level of privacy is controlled by the two positive parameters $\varepsilon$ and $\delta$; smaller values of these parameters correspond to stronger privacy protection. DP is also known for its particularly useful properties. (1) Post-processing: Any mapping or operation applied to the output of a differentially private mechanism will not compromise the privacy guarantees; (2) Composability: if all the components of a mechanism are differentially private, then their composition is also differentially private.

## 3.2  Diffusion Models

Diffusion models (DMs) are latent variable models trained to generate samples by gradually removing noise from data corrupted by Gaussian noise. The model learns a distribution $p_\theta(x_0)$ that approximates a data distribution $q(x_0)$. Diffusion models consist of two steps: the forward process and the reverse process. Let $X = \{x_0^{(n)}\}_{n=1}^N$ denote the empirical training set, and $x_0 \sim X$ as shorthand for sampling from its empirical distribution. The forward process gradually adds noise to the clean data sample $x_0 \sim X$, defined by the following Markov chain:

$$q(x_1, \ldots, x_T \mid x_0) = \prod_{t=1}^{T} q(x_t \mid x_{t-1}), \tag{1}$$

$$q(x_t \mid x_{t-1}) = \mathcal{N}(x_t; \sqrt{1 - \beta_t} x_{t-1}, \beta_t \mathbf{I}). \tag{2}$$

which is determined by a fixed noise variance schedule $\{\beta_1, \ldots, \beta_T\}$, where $\beta_t \in [0, 1]$ and $T$ is the total number of diffusion steps. The generation is performed by the reverse process, defined as the following Markov chain:

$$p_\theta(x_{0:T}) := p(x_T) \prod_{t=1}^{T} p_\theta(x_{t-1} \mid x_t), \tag{3}$$

$$p_\theta(x_{t-1} \mid x_t) := \mathcal{N}(x_{t-1}; \mu_\theta(x_t, t), \sigma_\theta(x_t, t)). \tag{4}$$

where $p(x_T) = \mathcal{N}(x_T; 0, \mathbf{I})$. We parameterize $\mu_\theta(x_t, t)$ and $\sigma_\theta(x_t, t))$ following the formulation of Ho et al. (2020). A deep neural network $\theta$ is trained to approximate the denoising function $\epsilon_\theta$ that predicts the noise $\epsilon$ from $x_t$, using the following loss function (where $x_t \sim q(x_t \mid x_0)$ via the forward process in equation 2):

$$\mathcal{L}_\theta = \mathbb{E}_{x_0 \sim X, \, \epsilon \sim \mathcal{N}(0, \mathbf{I}), t} \left[ \| \epsilon - \epsilon_\theta(x_t, t) \|^2 \right] \tag{5}$$

## 4 Problem Formulation

We consider each private multivariate time series as $x \in \mathbb{R}^{L \times F}$, where $L$ is the sequence length (horizon) and $F$ is the number of channels. Each times series sample represents a private entity, which is paired with conditional metadata $c$ representing public knowledge. While $c$ can be categorical or continuous in general, in this work we take $c$ to be time-varying metadata $c \in \mathbb{R}^{L \times K}$.

Let $D_{x,c} = (x_i, c_i)_{i=1}^n$ be a dataset consisting of $n$ independent and identically distributed samples, where each sample $(x_i, c_i)$ comprises a private multivariate time series $x_i \in \mathbb{R}^{L \times F}$ and its public metadata $c_i \in \mathbb{R}^{L \times K}$ (time points and channels within each $x_i$ are not i.i.d.).

Our goal is to learn a conditional generation model $G$, which generates data $D_{x'} = \{x' | x' = G(c)\}$ such that $p(x'|c) \approx p(x|c)$, and $G$ is $\varepsilon, \delta-$differentially private with respect to the sensitive component $D_x$.

## 5 Methodology

Here we propose Pub2Priv, a privacy-aware conditional diffusion framework for generating private data using public knowledge. Our model is trained with strong differential privacy (DP) guarantees through the application of DP-SGD (Abadi et al., 2016). While DP-SGD is a well-established method for deep learning with differential privacy, our goal is not to re-invent it. Instead, the novelty of our work lies in problem formulation itself and leveraging heterogeneous public data within the privacy-aware generation framework. Our model is composed of two main components, as illustrated in Figure 2: a conditional diffusion model $\theta_{\mathrm{DM}}$ that generates private time series data $x$, and a knowledge transformer $\theta_{\mathrm{T}}$ that creates temporal and feature embedding of public metadata $c$.

### 5.1 Public Knowledge Embedding

**CSDI-style encoder.** The first component of our model is a pre-trained transformer that produces embeddings from heterogeneous public metadata $c$. Following Tashiro et al. (2021); Narasimhan et al. (2024), we adopt a *two-dimensional* self-attention architecture that alternates attention along the temporal and feature axes to jointly model dynamics over time and dependencies across channels. Let $c \in \mathbb{R}^{k \times L}$ denote $k$ public signals observed over $L$ time steps. We first applies a **temporal** Transformer encoder to every channel independently (self-attention over $\{t = 1, \ldots, L\}$ at fixed $i$), and then a **feature** Transformer encoder to every time step independently (self-attention over $\{i = 1, \ldots, k\}$ at fixed $t$). This dual-axis factorization captures long-range temporal structure and inter-signal correlations while remaining memory-efficient for multivariate sequences. Variable-length sequences and missing entries are handled with attention masks as in Tashiro et al. (2021). The encoder output is a per-time embedding

$$z \;=\; \theta_{\mathrm{T}}(c) \;\in\; \mathbb{R}^{L \times d_{\mathrm{meta}}},$$

which conditions the downstream diffusion model.

**Positional encodings.** We use *2D positional signals* to distinguish order in time and features. In particular, we add a sinusoidal *temporal* position $p_t \in \mathbb{R}^d$, and a learned *feature* embedding $f_i \in \mathbb{R}^d$ to $c$ before providing it as input to $\theta_T$. The temporal encoder attends over $\{c_{i,1}, \ldots, c_{i,L}\}$ at fixed $i$, while the feature encoder attends over $\{c_{1,t}, \ldots, c_{k,t}\}$ at fixed $t$. For padding or irregular sampling, we supply a binary mask $m_{i,t}$ to both attention passes so that padded/missing positions neither attend nor are attended to.

**Pretraining objective for $\theta_{\mathrm{T}}$.** We pretrain $\theta_{\mathrm{T}}$ on publicly available time series data including stock prices and weather from Yahoo Finance and NCEI, using a *masked reconstruction* objective that encourages the encoder to capture both temporal and cross-channel structure. Let $M \in \{0,1\}^{k \times L}$ (with 1 indicating a masked entry) and let $\tilde{c} = c \odot (1 - M)$ be the masked input. We use a one-layer decoder $g_\phi$ to map encoder states to reconstructions $\hat{c} = g_\phi(\theta_{\mathrm{T}}(\tilde{c}))$. We minimize the per-feature normalized mean-squared error on the

masked set $\Omega = \{(i,t) : M_{i,t} = 1\}$:

$$\mathcal{L}_{\theta_T} = \frac{1}{|\Omega|} \sum_{(i,t) \in \Omega} \frac{\left\| \widehat{c}_{i,t} - c_{i,t} \right\|_2^2}{\sigma_i^2 + \epsilon},$$

where $\sigma_i^2$ is the variance of feature $i$ estimated on the public corpus (stabilized by $\epsilon > 0$). During pretraining we apply zero-padding and attention masking for variable-length sequences, with the same masks passed to both temporal and feature attention. After pretraining, $\theta_T$ is fixed and its outputs $z \in \mathbb{R}^{L \times d_{\mathrm{meta}}}$ serve as the conditioning signal for our diffusion model.

**Remark.** This design is inspired by CSDI and TimeWeaver of factorize attention along time and features to efficiently encode multivariate sequences (Tashiro et al., 2021; Narasimhan et al., 2024), while tailoring the objective to produce reusable public-context embeddings for conditioning.

## 5.2 Conditional Diffusion Model trained using DP-SGD

We employ the diffusion model as the backbone of Pub2Priv to provide generative capability. To utilize both private data $x$ and metadata embedding $z$, we condition the diffusion model on the public knowledge embedding to generate private time series data. Following the recent work on conditional diffusion models (Tashiro et al., 2021), we maintain the same forward process as in eq. (2), and define the reverse process as follows:

$$p_\theta(x_{t-1}|x_t, z) := \mathcal{N}(x_{t-1}; \mu_\theta(x_t, t|z), \sigma_\theta(x_t, t|z)) \tag{6}$$

This formulation provides the knowledge embedding during each diffusion step $t$ without any added noise. Given the denoiser $\theta_{\mathrm{DM}}$ and the knowledge transformer $\theta_T$, Pub2Priv minimize the following modified loss function:

$$\mathcal{L}_{\theta_{\mathrm{DM}}, \theta_T} = \mathbb{E}_{x \sim X, \epsilon \sim \mathcal{N}(0, \mathbf{I}), t} \left[ \| \epsilon - \theta_{\mathrm{DM}}(x_t, t|\theta_T(c)) \|^2 \right]. \tag{7}$$

We employ DP-SGD to protect the private data during training, which consists of two major procedures: gradient clipping and gradient noise addition. As illustrated in algorithm 1, DP-SGD randomly samples mini-batches $B_\tau$ from the training data with probability $\frac{B}{n}$ and computes the gradients (where $B$ is the size of the mini-batch). To bind the influence of each individual data point on model parameters, the gradients are clipped according to their $\ell_2$ norm and the clipping threshold $C$ (we explore $C \in \{0.1, 0.5, 1.0, 1.5, 2.0\}$ and select the one with lowest validation loss). After clipping, Gaussian noise $\mathcal{N}(0, \sigma^2 C^2 I)$ is added to the averaged gradient to protect privacy, and the model parameters are updated using the noisy gradients. The overall privacy loss is tracked throughout the training process using techniques such as the moments' accountant or Rényi Differential Privacy (RDP), ensuring that the cumulative privacy budget remains within $\varepsilon$ and $\delta$.

## 5.3 Privacy Analysis

According to Abadi et al. (2016), the generator $\theta_{DM}$ is differentially private for $D_x$ with the protection from gradeint noise and norms. Since $\theta_T$ is a pre-trained model that have no access the private data, it does not bring any privacy loss to the model. Therefore, $(\varepsilon, \delta)-$DP holds for our model (detail proof included in the appendix B).

# 6 Experiments

In this section, we assess the performance of Pub2Priv by analyzing the privacy-utility trade-off in comparison to state-of-the-art baselines across multiple domains.

---

**Algorithm 1** Training Algorithm for Differentially Private Generator $\theta_{\text{DM}}$

---

1: Initialize $\theta_{\text{DM}}$ randomly
2: **for** $\tau = 1$ to $N$ **do**
3:     Sample mini-batch $B_\tau$ with probability $\frac{B}{n}$
4:     **for** each $i \in B_\tau$ **do**
5:         $g_{\text{DM}}(x_i, c_i) \leftarrow \nabla_{\theta_{\text{DM}}} \mathcal{L}_{\theta_{\text{DM}}}(x_i, c_i)$
6:     **Gradient clipping:**
7:     $\bar{g}_{\text{DM}}(x_i, c_i) \leftarrow g_{\text{DM}}(x_i, c_i) / \max\left(1, \frac{\|g_{\text{DM}}(x_i, c_i)\|_2}{C}\right)$
8:     **Adding noise:**
9:     $\tilde{g}_{\text{DM}} \leftarrow \frac{1}{B}(\sum_{i \in B_\tau} \bar{g}_{\text{DM}}(x_i, c_i) + \mathcal{N}(0, \sigma^2 C^2 I))$
10:     **Model update:**
11:     $\theta_{\text{DM}} \leftarrow \theta_{\text{DM}} - \eta \cdot \tilde{g}_{\text{DM}}$

---

## 6.1 Baselines

Since there are no existing privacy-aware data generation model utilizing heterogeneous metadata, we adapt state-of-the-art DP generation models to incorporate metadata conditions alongside private inputs. In particular, we consider DP-GAN (Xie et al., 2018), which incorporates DP-SGD into the discriminator of the GAN framework, and PATE-GAN (Jordon et al., 2018), a GAN model leveraging Private Aggregation of Teacher Ensembles (PATE). These models are widely recognized as state-of-the-art approaches for integrating differential privacy into deep generative frameworks, focusing on minimizing per-sample reconstruction loss. In addition, we consider other leading methods for synthetic data generation that preserve marginal statistics, particularly for query release purposes. These include GEM (Liu et al., 2021), AIM (McKenna et al., 2022), and Private-GSD (Liu et al., 2023). Implementation details and model parameters of the baselines are shown in appendix E.

## 6.2 Datasets

While our model is broadly applicable to various data modalities, in this study we focus on three time series datasets from the finance, energy, and international trade domains.

### 6.2.1 Investment portfolios.

One of the most important secrets in financial activities is the investment portfolio, which represents the number of assets possessed by an investor. The changes in portfolios over time reveal private information of clients and their strategies. We consider the private data, which contains 1260 portfolio time series, each representing the daily holding positions of four unknown stock according to the contrarian strategy (Sharpe, 2010) or momentum strategy (Jegadeesh & Titman, 1993). We use the corresponding S&P500, Dow Jones industrial index, and common stock prices as public knowledge of the market (more information of the dataset can be found in appendix C).

### 6.2.2 Electricity usage.

The private data contains the daily electricity consumption of 370 users in Évora, Portugal (Bessa et al., 2015; Trindade, 2015) from 2011 to 2014. We utilize the daily average temperature and the monthly electricity price as public knowledge.

### 6.2.3 Semiconductor trading.

We also collected international trading data from the UN Comtrade dataset [1]. Specifically, we selected ten representative countries and collected their monthly import values in the electronic integrated circuit category from the year 2010 to 2023 as the private dataset. We collected the corresponding Philadelphia

---

[1]UN Comtrade dataset: https://comtradeplus.un.org/

Table 1: Utility of Pub2Priv and the baseline models ($\varepsilon = 1, \delta = 1 \times 10^{-5}$). For all metrics, smaller values indicate better utility. The best performances among Pub2Priv and the baselines are marked in bold text, excluding the ablation models.

| | Model | $KS_R$ | $KS_{AR}$ | $|Corr_{meta}|$ | TSTR (discri.) | TSTR (predic.) |
|---|---|---|---|---|---|---|
| | Pub2Priv | **0.28 ± 0.02** | **0.17 ± 0.05** | **0.18 ± 0.01** | **0.100 ± 0.08** | **0.003 ± 0.000** |
| | Pub2Priv(w/o $c$) | 0.43 ± 0.04 | 0.42 ± 0.02 | 0.53 ± 0.00 | 0.210 ± 0.065 | 0.008 ± 0.001 |
| | Pub2Priv(w/o $\theta_T$) | 0.42 ± 0.04 | 0.44 ± 0.02 | 0.48 ± 0.03 | 0.220 ± 0.057 | 0.016 ± 0.003 |
| Portfolio | DP-GAN | 0.29 ± 0.01 | 0.26 ± 0.01 | 0.38 ± 0.05 | 0.262 ± 0.065 | 0.006 ± 0.000 |
| | PATE-GAN | 0.46 ± 0.01 | 0.42 ± 0.02 | 0.52 ± 0.01 | 0.282 ± 0.160 | 0.054 ± 0.009 |
| | GEM | 0.36 ± 0.00 | 0.56 ± 0.00 | 0.53 ± 0.00 | 0.481 ± 0.033 | 0.053 ± 0.010 |
| | AIM | 0.40 ± 0.00 | 0.56 ± 0.01 | 0.53 ± 0.00 | 0.340 ± 0.163 | 0.008 ± 0.000 |
| | Private-GSD | 0.46 ± 0.00 | 0.56 ± 0.00 | 0.53 ± 0.00 | 0.391 ± 0.011 | 0.063 ± 0.004 |
| | Pub2Priv | 0.26 ± 0.01 | **0.17 ± 0.02** | **0.08 ± 0.02** | 0.119 ± 0.066 | **0.007 ± 0.007** |
| | Pub2Priv(w/o $c$) | 0.28 ± 0.01 | 0.32 ± 0.01 | 0.35 ± 0.00 | 0.155 ± 0.174 | 0.015 ± 0.001 |
| | Pub2Priv(w/o $\theta_T$) | 0.26 ± 0.01 | 0.18 ± 0.01 | 0.08 ± 0.02 | 0.121 ± 0.048 | 0.007 ± 0.005 |
| Electricity | DP-GAN | **0.25 ± 0.01** | 0.18 ± 0.03 | 0.10 ± 0.03 | **0.111 ± 0.145** | 0.012 ± 0.001 |
| | PATE-GAN | 0.29 ± 0.00 | 0.32 ± 0.01 | 0.35 ± 0.00 | 0.125 ± 0.161 | 0.042 ± 0.001 |
| | GEM | 0.34 ± 0.00 | 0.32 ± 0.00 | 0.34 ± 0.00 | 0.152 ± 0.140 | 0.054 ± 0.006 |
| | AIM | 0.25 ± 0.00 | 0.32 ± 0.00 | 0.34 ± 0.00 | 0.159 ± 0.204 | 0.049 ± 0.004 |
| | Private-GSD | 0.34 ± 0.00 | 0.32 ± 0.00 | 0.35 ± 0.00 | 0.324 ± 0.087 | 0.051 ± 0.006 |
| | Pub2Priv | **0.32 ± 0.03** | **0.77 ± 0.01** | **0.44 ± 0.05** | **0.113 ± 0.032** | **0.013 ± 0.001** |
| | Pub2Priv(w/o $c$) | 0.37 ± 0.00 | 0.77 ± 0.01 | 0.68 ± 0.01 | 0.332 ± 0.045 | 0.015 ± 0.002 |
| | Pub2Priv(w/o $\theta_T$) | 0.33 ± 0.03 | 0.77 ± 0.00 | 0.48 ± 0.04 | 0.111 ± 0.038 | 0.011 ± 0.001 |
| Comtrade | DP-GAN | **0.32 ± 0.00** | 0.79 ± 0.00 | 0.59 ± 0.06 | 0.303 ± 0.117 | **0.013 ± 0.002** |
| | PATE-GAN | 0.33 ± 0.00 | 0.79 ± 0.00 | 0.68 ± 0.01 | 0.290 ± 0.148 | 0.046 ± 0.005 |
| | GEM | 0.19 ± 0.00 | 0.26 ± 0.00 | 0.33 ± 0.00 | 0.215 ± 0.145 | 0.073 ± 0.008 |
| | AIM | 0.20 ± 0.00 | 0.25 ± 0.00 | 0.32 ± 0.00 | 0.350 ± 0.042 | 0.044 ± 0.003 |
| | Private-GSD | 0.22 ± 0.00 | 0.26 ± 0.00 | 0.33 ± 0.01 | 0.275 ± 0.225 | 0.061 ± 0.009 |

semiconductor sector index (SOX) time series, using Yahoo Finance Python API [2], as the public data since the index values are highly correlated to the semiconductor import volume.

## 6.3 Utility Evaluation

### 6.3.1 Time series utility metrics

We consider five metrics for assessing the utility of generated time series: $KS_R$ and $KS_{AR}$ (Kolmogorov–Smirnov statistics of *return* and *autocorrelation* distributions, respectively); $|Corr_{meta}|$, the absolute gap between real and synthetic metadata–data correlations; and two Train on Synthetic Test on Real scores TSTR *discriminative* and *predictive*. Formal definitions, aggregation rules, and implementation details are provided in App. F.

The discriminative score measures how well an RNN-based discriminator can distinguish between original and synthetic time series data samples. The original and synthetic data are evenly distributed in both training and testing datasets. The discriminative score is defined as the absolute difference between the testing accuracy and 0.5, which is the probability of a random guess. Thus, a score close to 0 indicates that the synthetic data is indistinguishable from the real data. Conversely, a score farther from 0 implies that synthetic data samples are less realistic and significantly different from real data samples.

The predictive score accesses how well the synthetic data captures the underlying patterns and dynamics in the real data. All synthetic data is utilized to train an RNN-based predictor, and all real data is used for testing. The predictive score is the mean square error between the predicted values and the real values.

---

[2] yfinance Python package: https://pypi.org/project/yfinance/

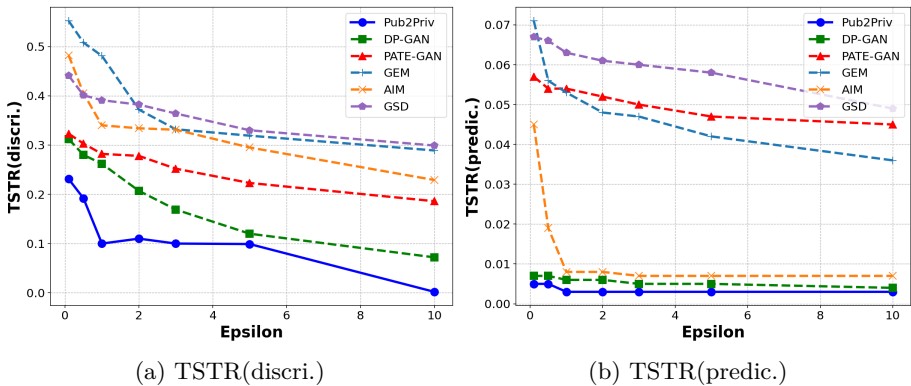

(a) TSTR(discri.)            (b) TSTR(predic.)

Figure 3: The utility-privacy trade-off for Pub2Priv and the benchmark models on the portfolio dataset.

A lower score indicates that synthetic data samples capture important patterns and features in real data, making them useful in developing predictive models.

### 6.3.2 Experimental results

We compare our model to baselines under the same privacy budget ($\varepsilon = 1$, $\delta = 1 \times 10^{-5}$). Each experiment is repeated 10 times; we report mean $\pm$ standard deviation in table 1. Formal metric definitions and aggregation rules are in App. F.

**Distributional fidelity.** As shown in table 1, our model achieves the best performance in preserving the return distribution on the Portfolio dataset and attains better or comparable $KS_R$ on the other two datasets. For autocorrelations, our method consistently outperforms all baselines across datasets in $KS_{AR}$.

**Private–public correlations.** We report $|\mathrm{Corr}_{\mathrm{meta}}|$ to assess how well synthetic data reproduce correlations between private data $x$ and public metadata $c$. Note that while our model is specifically designed to extract and utilize knowledge from heterogeneous metadata, $|\mathrm{Corr}_{\mathrm{meta}}|$ is a diagnostic metric which is not included in the training objectives of $\theta_T$ or the diffusion model $\theta_{DM}$. Our model yields the smallest $|\mathrm{Corr}_{\mathrm{meta}}|$ across datasets, indicating better preservation of private–public dependencies (table 1).

**Downstream utility (TSTR).** Our approach delivers superior TSTR performance on average: the discriminative score improves by 0.118 (higher is better) and the predictive score by 0.013 (lower MSE is better) compared to baselines (table 1). DP–GAN achieves the best discriminative TSTR on the Electricity dataset but exhibits high variance (standard deviation larger than the mean), indicating instability.

**Privacy–utility trade-off.** Across privacy budgets, our method consistently yields better TSTR scores than baselines (fig. 3). When the budget is very tight, all methods struggle; as $\varepsilon \to 1$, our model increasingly captures useful structure tied to the public signals, improving utility. We observed that the marginal-statistics-based baselines (GEM, AIM, and Private-GSD) perform poorly in terms of TSTR scores. This is primarily because these methods are designed to generate synthetic data that supports accurate responses to aggregate queries, rather than to capture realistic individual-level patterns. Consequently, they fail to produce synthetic samples that reflect plausible trajectories or behaviors, such as realistic investment portfolios. As a result, their TSTR performance is significantly lower due to the lack of individual-level fidelity.

**Qualitative distributional coverage.** In addition to the quantitative evaluations, we use t-SNE (Van der Maaten & Hinton, 2008) plots to visualize the distributions of the synthetic and real data. Figure 4 shows that DP-GAN, PATE-GAN, and AIM deviate substantially from the real distribution under $\varepsilon = 1$, $\delta = 1 \times 10^{-5}$. GEM and Private–GSD tend to cluster around a few outliers rather than covering the bulk of the data. In contrast, our samples more closely match the real distribution.

**Summary.** Overall, our model surpasses baselines on most metrics and datasets and improves the privacy–utility trade-off (see table 1 and fig. 3).

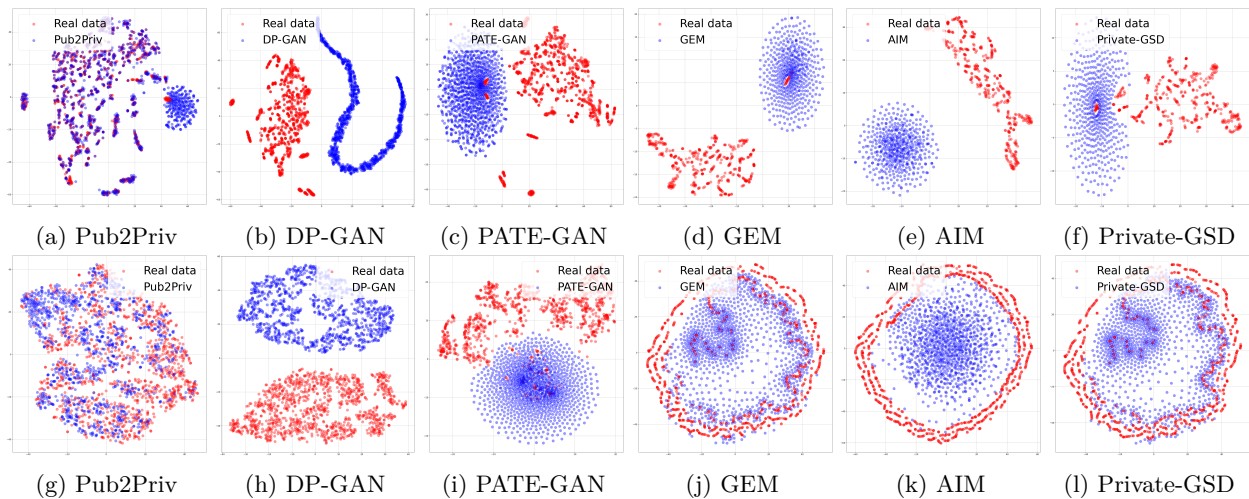

Figure 4: t-SNE visualizations of synthetic data generated by Pub2Priv and the baseline models based on $\varepsilon = 1, \delta = 1 \times 10^{-5}$, where the top row shows portfolio dataset and the bottom are electricity dataset. We omit the t-SNE plots for the Comtrade dataset due to its relatively small size, which results in sparse visualizations.

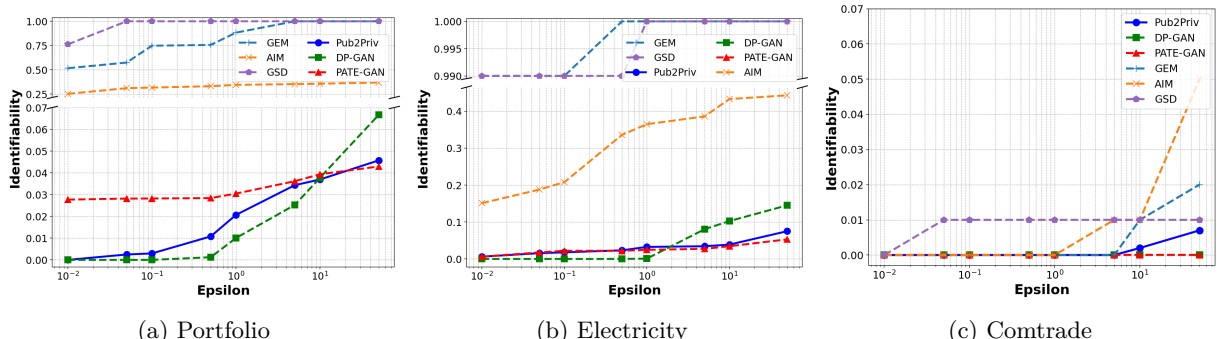

Figure 5: The identifiability $\mathcal{I}(D, D')$ of synthetic data generated by Pub2Pub and the benchmark models given different $\varepsilon$ (with $\delta = 1 \times 10^{-5}$).

### 6.4 Empirical Identifiability Evaluation

While DP-SGD provides a strong $(\varepsilon, \delta)-$DP guarantee to our model training, we consider a practical privacy metrics that allow us to compare with models trained with other privacy mechanisms. Similar to Yoon et al. (2020), we propose the synthetic data identifiability, which is defined as:

$$\mathcal{I}(D, D') = \frac{1}{n} \sum |x'_i \in D'; d_i < d'_i| \tag{8}$$

where $D$ and $D'$ are the real and synthetic data. For a synthetic sample $x'_i \in D'$, $d_i = \min_{x_j \in D} ||x'_i - x_j||$ represents the minimum distance between $x'_i$ and all samples in the real data, whereas $d'_i = \min_{x'_j \in D' \setminus x'_i} ||x'_i - x'_j||$ denotes the minimum distance to all other synthetic data samples. If $d_i < d'_i$, $x'_i$ is closer to real data $x_j \in D$ than any other synthetic data points, posing a potential risk of exposing the information of $x_j$. The identifiability $\mathcal{I}(D, D')$ represents the proportion of synthetic data that might be "identifiable" to the real data, where less identifiability indicates stronger privacy.

Now we evaluate the practical identifiability $\mathcal{I}(D, D')$ of the synthetic data generated by our model. Figure 5 shows that our model yields synthetic data with a similar level of identifiability compared to the baseline

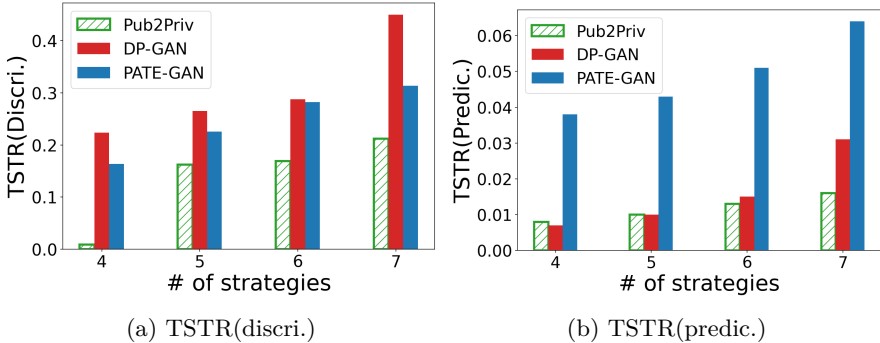

Figure 6: The utility of synthetic data generated by Pub2Priv and the benchmark models given private portfolios of different complexity ($\varepsilon = 1$, $\delta = 1 \times 10^{-5}$). As the number of strategies increases, the portfolios are more diverse and the complexity of public-private data relations increases. TSTR scores from the three marginal statistics based methods (GEM, AIM, Private-GSD) are omitted as they are significantly worse than the above.

models. For all datasets, including portfolio, electricity, and Comtrade, the synthetic data generated by Pub2Priv has $\mathcal{I}(D, D') \leq 0.04$ when $\varepsilon = 1$, meaning that less than 4% of the synthetic time series is closer to the real private data. We also observe that $\mathcal{I}(D, D')$ and $\varepsilon$ are positively correlated in most cases, which indicates that identifiability can be considered as an alternative way to evaluate and configure the privacy of generation models. However, it's important to note that $\varepsilon$ alone does not guarantee low identifiability. For example, consider a model that generates synthetic data concentrated around the geometric center of the training distribution. Such a model may yield a low $\varepsilon$ (since individual points have minimal influence), yet still result in high identifiability loss—especially if there is a real individual near the center. This kind of behavior can be seen in the t-SNE plots of PATE-GAN, GEM, and Private-GSD, which emphasize the importance of evaluating empirical privacy leakage (e.g., identifiability) in addition to enforcing theoretical DP guarantees.

## 6.5 Impact of public-private interconnection

Our method is particularly effective when there is a correlation between public and private data. To further assess its robustness, we investigate performance changes as the private data to be generated becomes more complex while the public metadata given remains the same. Specifically, given the same public metadata, we consider private data comprising various portfolio strategies and evaluate our model's effectiveness (details of these strategies are provided in appendix C.1). Since investments in the same set of assets can yield different portfolios depending on the trading strategy used, private data containing a large number of strategies tends to have a more indirect and loosely defined relationship with the public metadata (i.e., stock prices). As a result, inferring private data from public knowledge becomes significantly more challenging. Figure 6 presents the TSTR scores when varying the number of strategies in the Portfolio dataset (public metadata held fixed). As the number of strategies increases, the diversity and complexity of the portfolio dataset grows and we observe a general decline in performance across methods within this controlled setting. However, while the baselines exhibit a much steeper rise in TSTR loss, our model more effectively captures the intricate and diverse public–private interconnections, resulting in a more gradual increase in TSTR loss. We further evaluate the impact by controlling the public-private data correlations and show the results in appendix G.1.

## 6.6 Ablation Study

The superior performance of Pub2Priv comes from two key components: (1) the use of public metadata and (2) the knowledge transformer $\theta_T$ that incorporates the temporal and feature embedding of public knowledge. We conduct the ablation study to investigate the effectiveness of these two modules. The first ablation method is Pub2Priv(w/o $c$), which is simply our generator without giving any public metadata. The second ablation method Pub2Priv(w/o $\theta_T$) is created by removing $\theta_T$ and feeding $c$ directly to the denoiser

$\theta_{\mathrm{DM}}$ as conditional input. Table 1 represents the utility of synthetic data generated by Pub2Priv(w/o $c$) and Pub2Priv(w/o $\theta_{\mathrm{T}}$). The results indicate that the TSTR scores of Pub2Priv(no w/o $c$) are 15% to 194% worse than those of the original model across all datasets, highlighting the benefit of utilizing public metadata. Although performance degradation is also observed in Pub2Priv(w/o $\theta_{\mathrm{T}}$), the effectiveness of $\theta_{\mathrm{T}}$ is diminished for datasets with lower dimensionality and less informative public data.While performance declines are also observed for Pub2Priv(w/o $\theta_{\mathrm{T}}$), the $\theta_{\mathrm{T}}$ is less effective for datasets with lower dimensionality and a smaller amount of public information (electricity and comtrade datasets). Overall, the ablation study demonstrates that both the public metadaa and knowledge transformer $\theta_{\mathrm{T}}$ are essential components of Pub2Priv, and their removal significantly weakens the model's generative capabilities.

## 7 Limitations and Discussion

While our model has shown promising results in enhancing the privacy-utility trade-off, it represents only an initial step in the broader exploration of leveraging heterogeneous public knowledge for privacy-aware data generation. In this study, the selection of public metadata was guided by discretion on the following three factors:

**Utility.** Like other studies in semi-private learning (Alon et al., 2019; Lowy et al., 2023; Wang & Zhou, 2020), our method relies on the assumption that public data is both relevant and useful. Experimental results indicate that our approach benefits more from richer public metadata with higher dimensionality (e.g., portfolio data) compared to lower-dimensional sources (e.g., electricity and Comtrade data). If the private data distribution becomes more complex (larger variety of strategies in section 6.5), a larger amount of public metadata may be necessary to preserve the utility of the generated synthetic data.

**Safety.** We exclusively consider non-sensitive contextual information that does not reveal any individual details from the private data. For instance, market indices used as public data reflect overall market conditions without disclosing specific portfolio compositions or strategies. Likewise, semiconductor stock prices as public data are not tied to any particular country. To mitigate this restriction, future research could explore additional mechanisms for incorporating sensitive metadata in a privacy-preserving manner.

**Availability.** While our approach expands on existing semi-private learning research by leveraging heterogeneous public metadata rather than solely homogeneous public data, identifying relevant public information still requires domain expertise. To overcome this limitation, future work could harness the empirical knowledge embedded in large language models (LLMs) to automatically identify useful public datasets (Zhu et al., 2024).

## 8 Conclusion

In this study, we introduce a conditional diffusion framework for privacy-aware time series generation that leverages heterogeneous public knowledge. Our model incorporates a self-attention mechanism to capture the temporal and feature correlations in the heterogeneous metadata, and employs DP-SGD to protect data privacy. Experiment evaluations show that given the same privacy budget, our model generates time series with better privacy-utility trade-off for datasets in various domains. Ablation studies validate the importance of the use of public metatdata and the knowledge transformer.

## Acknowledgments

We thank Elizabeth Fons and Vamsi K. Potluru for their thoughtful feedback and support.

## Disclaimer

or reliability of the information contained herein. This document is not intended as investment research or investment advice, or a recommendation, offer or solicitation for the purchase or sale of any security, financial instrument, financial product or service, or to be used in any way for evaluating the merits of participating in any transaction, and shall not constitute a solicitation under any jurisdiction or to any person, if such solicitation under such jurisdiction or to such person would be unlawful.

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

## A  Broader Impact

Our paper brings a novel problem formulation and introduces the first approach of privacy-aware data generation by leveraging public contextual information. We believe this is an important research area which extends semi-private learning to heterogeneous public data from non-sensitive sources and domains. For instance:

- In domains such as finance, generating synthetic portfolios and trading transactions enables researchers and analysts to develop and validate trading strategies without risking disclosing sensitive market positions. In this case, non-sensitive market indices and common stock prices can be used to enhance the generation of private portfolios.

- In healthcare, synthetic patient data can facilitate disease progression and resource allocation research without exposing individuals' private health records. Public available metadata like the spread of diseases and distribution of vaccine can be utilized.

- In energy domain, synthetic electrical consumption patterns can assist in smart grid simulations and forecasting while safeguarding the identities of households, which can benefit from the public knowledge of weather conditions and electricity pricing.

## B  Proof of Differential Privacy in Pub2Priv

In this section, we provide a concise proof that the gradients in Pub2Priv are differentially private (DP). Specifically, we consider the definition of Rényi Differential Privacy (RDP) from Mironov (2017):

**Definition Rényi Differential Privacy** A randomized mechanism $\mathcal{M} : \mathcal{D} \to \mathcal{R}$ with domain $\mathcal{D}$ and range $\mathcal{R}$ satisfies $(\alpha, \varepsilon)$-RDP if for any adjacent $D, D' \in D$:

$$D_\alpha(M(d) \mid M(d')) \leq \varepsilon, \tag{9}$$

where $D_\alpha$ is the Rényi divergence of order $\alpha$. Any $\mathcal{M}$ that satisfies $(\alpha, \epsilon)$-RDP also satisfies $(\epsilon + \log \frac{1}{\delta}/(\alpha - 1), \delta)$-DP.

**Theorem B.1.** *(Mironov, 2017) For a query function $f$ with Sensitivity $S = \max_{d,d'} \|f(d) - f(d')\|_2$, the Gaussian mechanism that releases $f(d) + \mathcal{N}(0, \sigma^2)$ satisfies $(\alpha, \alpha S^2/(2\sigma^2))$-RDP.*

In each iteration of Algorithm 1, we randomly sample a mini-batch $B_\tau$ with expected size $B$ with no repeated indices. We implement our model based on Yousefpour et al. (2021) which applies the Gaussian mechanism for gradient sanitization. After computing the gradient of $\mathcal{L}(x_i, z_i))$, we apply clipping with norm $C$, and then divide the clipped gradients by the expected batch size $B$ to obtain the batched gradient $G_{\mathrm{DM}}$:

$$G_{\mathrm{DM}}(\{x_i, z_i\}) = \frac{1}{|B|} \sum_{i \in B} \mathrm{clip}_C(\nabla_{\theta_{\mathrm{DM}}} \mathcal{L}(x_i, z_i)). \tag{10}$$

Finally, Gaussian noise $\epsilon \sim \mathcal{N}(0, \sigma^2)$ is added to $G_{\mathrm{DM}}$ and released as the response $\tilde{G}_{\mathrm{DM}}$:

$$\tilde{G}_{\mathrm{DM}}(\{x_i, z_i\}_{i \in B}) = G_{\mathrm{DM}}(\{x_i, z_i\}_{i \in B}) + \frac{C}{B}\epsilon. \tag{11}$$

Since $z_i$ is metadata embedding of $c_i$ which is free of privacy concerns, we can restate Theorem 2 as follow:

**Theorem B.2.** *For noise magnitude $\sigma_{DP}$, dataset $d = \{x_i\}_{i=1}^N$, and a set of samples $B_\tau$, releasing $\tilde{G}_{DM}(\{x_i\}_{i \in B})$ satisfies $(\alpha, \alpha/2\sigma^2)$-RDP.*

*Proof.* Considering two neighboring datasets $d = \{x_i\}_{i=1}^N$ and $d' = d \cup \{x'\}$, where $x' \notin d$, and mini-batches $\{x_i\}_{i \in B}$ and $\{x'\} \cup \{x_i\}_{i \in B}$, that differs by one additional entry $x'$. We can bound the difference of their gradients in $L_2$-norm as

| Dataset | # of Samples | Length | Pearson$(x,c)$ | Private Data | Public Data |
|---|---|---|---|---|---|
| Portfolio | 1260 | 360 | 0.573 | Daily hoding positions | Market indices and common stocks |
| Electriciy | 1404 | 365 | 0.386 | Electriciy usage | Daily average temperature; monthly electricity price |
| Comtrade | 100 | 120 | 0.761 | Monthly trading values | Semiconductor index (SOX) |

Table 2: Summary of the datasets.

$$\|G_{\mathrm{DM}}(\{x_i\}_{i\in B}) - G_{\mathrm{DM}}(\{x'\}\cup\{x_i\}_{i\in B})\|_2$$

$$= \left\|\frac{1}{B}\sum_{i\in B}\mathrm{clip}_C(\nabla_\theta\mathcal{L}(x_i)) - \frac{1}{B}\left(\mathrm{clip}_C(\nabla_\theta\mathcal{L}(x')) \ -\sum_{i\in B}\mathrm{clip}_C(\nabla_\theta\mathcal{L}(x_i))\right)\right\|_2$$

$$= \frac{1}{B}\|\mathrm{clip}_C(\nabla_\theta\mathcal{L}(x'))\|_2 \le \frac{C}{B}. \tag{12}$$

This difference is bounded by the sensitivity of $\frac{C}{B}$, which is accounted for in the Gaussian mechanism under Mironov (2017). Furthermore, since $\epsilon\sim\mathcal{N}(0,\sigma^2)$, it follows that $\frac{C}{B}\epsilon\sim\mathcal{N}\left(0,\left(\frac{C}{B}\right)^2\sigma^2\right)$. Following standard arguments, releasing $\tilde{G}_{\mathrm{DM}}(\{x_i\}_{i\in B}) = G_{\mathrm{DM}}(\{x_i\}_{i\in B}) + \frac{C}{B}\epsilon$ satisfies $(\alpha,\alpha/2\sigma^2)$-RDP. Since $c$ is not private data, $\theta_{\mathrm{T}}$ does not bring additional privacy cost by the post-processing property of differential privacy (Dwork et al., 2014). In practice, we construct mini-batches by sampling the training dataset for privacy amplification via Poisson Sampling (Mironov et al., 2019). The overall privacy cost of training $\theta_{\mathrm{DM}}$ is computed via RDP composition (Mironov, 2017), using the processes implemented in Opacus (Yousefpour et al., 2021).

## C  Dataset Description

In this section, we describe the details of the datasets used for our experiment evaluation, including the private time series data and public domain knowledge. Table 2 presents the summary statistics of all datasets.

### C.1  Portfolio Dataset

We consider the classic investment return maximization problem (Cover, 1984) which allocates investment capital over the stocks. The private data presents the amount of holdings on each day, which contains 1260 portfolio time series created based on the following two strategies:

#### C.1.1  Contrarian Strategy

The contrarian trading strategy is introduced by Sharpe (2010), which represents an adaptive asset allocation policy. An investor starts with an initial portfolio of value $V_0 = \sum_i X_i$ where $X_i$ is the amount of money invested in asset $i$. At each reviewing period $t$, the investor adjust the proportion of assets by adding the adjusting value $D_i$ (purchase if positive and sell if negative) for each assets:

$$D_i = (K_p - k_i)X_i \tag{13}$$

where $K_p = \frac{V_t}{V_0}$ and $k_i$ is the return of the asset.

#### C.1.2  Momentum Strategy

The momentum strategy is based on the principle that stocks that have performed well in the past will continue to perform well in the future, while stocks that have performed poorly will continue to underperform. At each reviewing period $t$, it set the invested value $X_i = k_i V_t$ for each asset.

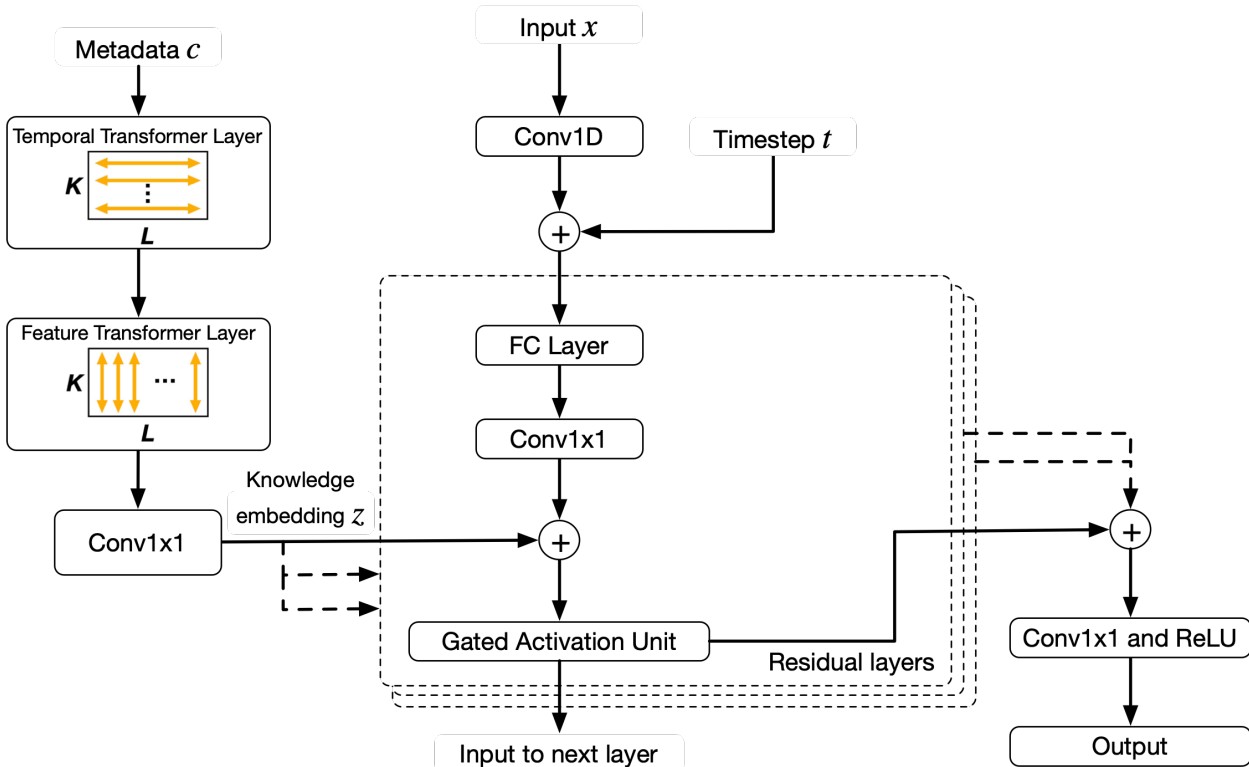

Figure 7: Pub2Priv architecture. We use self-attention layers $\theta_{\mathrm{T}}$ to create temporal and feature embedding of the metadata $c$, which is passed to every residual layers of the denoiser $\theta_{\mathrm{DM}}$ to generate the output for each denoising step $t$.

Note that multiple strategies can be derived from the above two policies by considering different length of reviewing period for calculating asset returns. To protect investor identity and their business strategies, each portfolios was built on an unknown stocks from the Dow Jones composite components during an unknown time period of 360 consecutive days excluding weekends. For public knowledge, we consider the Standard and Poor's 500 index (S&P 500), which is a stock market index tracking the stock performance of 500 of the largest publicly traded companies on stock exchanges in the United States. We also consider the Dow Jones Industrial index and the price of common stocks including AAPL, AMZN, MSFT, NVDA, and TSLA.

## C.2 Electricity Dataset

The electricity dataset contains power consumption recorded for 370 users over a period of 4 years from 2011 to 2015 provided by Trindade (2015). Daily usage are created by aggregating the 15 minutes power consumption in the raw data. As all users are located in Évora, Portugal, we randomly selected four 1-year samples for each user in order to create a dataset for conditionally generating time series based on specific public metadata. This results in 1404 time series as some users have consumption record shorter or equal to one year. We consider the daily average temperature and the monthly electricity price as public knowledge.

## C.3 Comtrade Dataset

We also collected time series data from the UN Comtrade data source, which is a comprehensive and widely-used data resource for international trading statics. This dataset provides data recorded by the United Nations about many aspects in global trading, including information on imports and exports, trading categories, trading values and volumes. We specifically focused on the import values of the electronic integrated circuits category, which category code is 8542, from 10 selected countries (Spain, USA, Germany, Japan, United Kingdom, France, Brazil, Italy, Canada, Australia). We collected the corresponding monthly trad-

ing values from Jan 2010 to Dec 2023. For each country, we sampled 10 time series samples with random starting months, and we set the length of samples to 120. Thus, we collected 100 time series samples with each length of 120 as the private dataset for our utility studies. The corresponding public data was the Philadelphia semiconductor sector index (SOX) monthly time series collected via the Yahoo Finance Python API. Similarly, the SOX samples were also collected with the length of 120. So that the public dataset share the same data quantity as the private dataset.

## D Pub2Priv Model Architecture

We present additional details about the Pub2Priv architetcture in order to improve the reproducibility and usage of our framework. Figure 7 shows the architecture of the knowledge transformer $\theta_{DM}$ and denoiser $\theta_T$. Inspired by Tashiro et al. (2021), we use the self-attention layers to capture the temporal and feature correlations of the metadata $c$. We implemented the denoiser network $\theta_{DM}$ using residual layers similar to Narasimhan et al. (2024), which utilize the public knowledge embedding $z$ at each step to mitagate the loss from DP-SGD. Our model parameters are listed in table 3.

| Parameter | Value |
|---|---|
| $\theta_T$ embedding size | 128 |
| $\theta_T$ attention heads | 8 |
| $\theta_T$ self-attention layers | 8 |
| $\theta_T$ dropout | 0.05 |
| $\theta_T$ activation | GELU |
| $z$ size | 128 |
| # of residual layers | 4 |
| $\theta_{DM}$ hidden dimension | 256 |
| # of diffusion steps | 400 |
| $\beta_1$ | 0.0001 |
| $\beta_T$ | 0.2 |
| Batch size | 128 (port. & elec.); 16 (Comtrade) |
| Learning rate | $1 \times 10^{-4}$ |

Table 3: Hyperparamters of Pub2Priv.

| Parameter | Value |
|---|---|
| Generator layers | 4 |
| Generator hidden dimensions | 256 |
| Discriminator layers | 3 |
| Discriminator hidden dimensions | 128 |
| Discriminator iterations (DP-GAN) | 20 |
| # of teachers (PATE-GAN) | 10 |
| noise ratio (PATE-GAN) | 1.0 |
| noise size (PATE-GAN) | 1.0 |
| $\lambda$ (ADS-GAN) | 1.0 |
| Batch size | 128 (port. & elec.); 16 (Comtrade) |
| Learning rate | $1 \times 10^{-4}$ |

Table 4: Hyperparameters of baseline models.

## E Baseline Model Implementations

Here we report the hyerparameter configurations for the baseline models in table 4. We use the implementation provided by the original authors for all baseline methods: DP-GAN (Xie et al., 2018), PATE-GAN (Jordon et al., 2018), and ADS-GAN (Yoon et al., 2020). We concatenate the metadata as conditional input and add 1D convolution layer to adapt the baselines models for time series data. In addition, we adjusted the number of parameters in the generator and discriminator to roughly match the Pub2Priv models. Preserving marginal statistics has proven effective for generating synthetic data with privacy guarantees, particularly in the context of query release. However, these methods primarily target tabular data, focusing on queries such as the average age of individuals earning above \$20k. To adapt state-of-the-art marginal statistics–based approaches (GEM, AIM, and Private-GSD) to time series data, we apply an aggressive preprocessing strategy. Specifically, we divide each time series into 10 temporal segments and quantize the feature values into 50 discrete levels. This flattening process allows us to approximate the data as a 2D table for the query workloads, but at the cost of significant loss in temporal resolution and structure. We acknowledge that this approach is a crude and lossy adaptation, and indeed it confirms our concern: methods designed for preserving marginal statistics in static tabular data are not suitable for generating temporally coherent, realistic synthetic trajectories.

## F   Evaluation Metrics

Let $\mathcal{D}_{\text{real}} = \{x^{(n)} \in \mathbb{R}^{d \times L}\}_{n=1}^{N}$ be the set of real private time series (with $d$ channels and length $L$), $\mathcal{D}_{\text{syn}} = \{x'^{(n)} \in \mathbb{R}^{d \times L}\}_{n=1}^{N}$ the corresponding synthetic set, and $c^{(n)} \in \mathbb{R}^{k \times L}$ the public metadata. Unless noted, all computations are performed per sequence and per channel, then aggregated as specified.

**$\text{KS}_{\mathbf{R}}$: KS distance of return distributions.**   For each channel $i$ and sequence $n$, define first-difference returns

$$r_{i,t}^{(n)} = x_{i,t}^{(n)} - x_{i,t-1}^{(n)}, \quad r_{i,t}'^{(n)} = x_{i,t}'^{(n)} - x_{i,t-1}'^{(n)}, \qquad t = 2, \dots, L.$$

Let $F_i(u)$ and $F_i'(u)$ be the empirical CDFs of $\{r_{i,t}^{(n)}\}_{n,t}$ and $\{r_{i,t}'^{(n)}\}_{n,t}$, obtained by pooling over all $n$ and $t$. The per-channel KS statistic is

$$\text{KS}_R(i) = \sup_{u \in \mathbb{R}} \big| F_i(u) - F_i'(u) \big|.$$

We report the average across channels:

$$\text{KS}_R = \frac{1}{d} \sum_{i=1}^{d} \text{KS}_R(i).$$

**$\text{KS}_{\mathbf{AR}}$: KS distance of auto-correlation distributions.**   Fix a set of lags $\Lambda = \{1, \dots, L_{\max}\}$. For each channel $i$, sequence $n$, and lag $\ell \in \Lambda$, define the sample auto-correlation

$$\rho_{i,\ell}^{(n)} = \frac{\sum_{t=\ell+1}^{L} \left(x_{i,t}^{(n)} - \bar{x}_i^{(n)}\right)\left(x_{i,t-\ell}^{(n)} - \bar{x}_i^{(n)}\right)}{\sum_{t=1}^{L} \left(x_{i,t}^{(n)} - \bar{x}_i^{(n)}\right)^2}, \qquad \bar{x}_i^{(n)} = \frac{1}{L} \sum_{t=1}^{L} x_{i,t}^{(n)},$$

and similarly $\rho_{i,\ell}'^{(n)}$ from $x'^{(n)}$. For each lag $\ell$, let $G_\ell(u)$ and $G_\ell'(u)$ be the empirical CDFs of $\{\rho_{i,\ell}^{(n)}\}_{i,n}$ and $\{\rho_{i,\ell}'^{(n)}\}_{i,n}$. The lag-wise KS statistic is

$$\text{KS}_{AR}(\ell) = \sup_{u \in [-1,1]} \big| G_\ell(u) - G_\ell'(u) \big|,$$

and we aggregate over lags by averaging:

$$\text{KS}_{AR} = \frac{1}{|\Lambda|} \sum_{\ell \in \Lambda} \text{KS}_{AR}(\ell).$$

**$|\text{Corr}_{\mathbf{meta}}|$: metadata–data correlation gap.**   For each private channel $i \in \{1, \dots, d\}$, metadata channel $j \in \{1, \dots, k\}$, and sequence $n$, compute the Pearson correlation across time

$$\text{corr}_{i,j}^{(n)} = \text{Pearson}\big(x_{i,1:L}^{(n)}, c_{j,1:L}^{(n)}\big), \qquad \text{corr}_{i,j}'^{(n)} = \text{Pearson}\big(x_{i,1:L}'^{(n)}, c_{j,1:L}^{(n)}\big).$$

The absolute correlation gap is then averaged over all triplets:

$$\big|\text{Corr}_{\text{meta}}\big| = \frac{1}{Ndk} \sum_{n=1}^{N} \sum_{i=1}^{d} \sum_{j=1}^{k} \big| \text{corr}_{i,j}^{(n)} - \text{corr}_{i,j}'^{(n)} \big|.$$

**TSTR (discriminative).**   Given a supervised discriminative task with real labels $\{y^{(n)}\}$ (e.g., classification), train a predictor $h_\theta$ on $\{(x'^{(n)}, y^{(n)})\}$ using the synthetic inputs, then evaluate on a held-out real test set $\mathcal{T}_{\text{real}}$:

$$\theta_{\text{syn}}^{\star} = \arg\min_{\theta} \mathcal{L}_{\text{disc}}\big(h_\theta; \mathcal{D}_{\text{syn}}\big), \qquad \text{TSTR}_{\text{disc}} = \mathcal{M}_{\text{disc}}\big(h_{\theta_{\text{syn}}^{\star}}; \mathcal{T}_{\text{real}}\big),$$

where $\mathcal{L}_{\text{disc}}$ is a standard discriminative loss (e.g., cross-entropy) and $\mathcal{M}_{\text{disc}}$ is the target metric (e.g., accuracy or AUROC). Higher is better for accuracy/AUROC.

| Correlation | 0.0 | 0.3 | 0.6 | 0.9 |
|---|---|---|---|---|
| TSTR(discri) | 0.480 | 0.385 | 0.450 | 0.480 |
| TSTR(predic) | 0.022 | 0.020 | 0.015 | 0.005 |

Table 5: The TSTR performance of Pub2Priv ($\varepsilon = 1, \delta = 1 \times 10^{-5}$) on the same toy private dataset given public metadata with different degrees of public-private correlation.

**TSTR (predictive).** For a forecasting task with horizon $\Delta$ and window size $w$, construct supervised pairs from sliding windows. Train a forecaster $f_\theta$ on synthetic pairs and evaluate mean squared error on real test pairs:

$$\theta^\star_{\text{syn}} = \arg\min_\theta \frac{1}{|\mathcal{S}_{\text{syn}}|} \sum_{(X',Y') \in \mathcal{S}_{\text{syn}}} \left\| f_\theta(X') - Y' \right\|_2^2, \quad \text{TSTR}_{\text{pred}} = \frac{1}{|\mathcal{S}_{\text{real}}|} \sum_{(X,Y) \in \mathcal{S}_{\text{real}}} \left\| f_{\theta^\star_{\text{syn}}}(X) - Y \right\|_2^2.$$

Here each $X \in \mathbb{R}^{d \times w}$ is a length-$w$ window from a real series and $Y \in \mathbb{R}^{d \times \Delta}$ is the $\Delta$-step-ahead target; $(X', Y')$ are constructed analogously from synthetic series. Lower is better for MSE. We report means over multiple random seeds/splits.

## G  Additional Experiment Results

In this section, we present additional experiments that do not fit into the main body of the paper.

### G.1  Impact of public-private interconnection

In addition to the portfolio experiment in section 6.5, we have included a new toy scenario to provide a more intuitive understanding of how the correlation between public and private data affects our model. Specifically, we construct a toy dataset of 100 one-dimensional time series, where each private series $x_i$ is composed of trend, seasonality, and small noise components. Corresponding public metadata $c_i$ is generated as $c_i = \alpha x_i + \sqrt{1 - \alpha^2} \cdot N(\mu, \sigma)$ where $\alpha$ is the desired Pearson correlation. As shown in the table 5, we observe that stronger correlation improves the TSTR predictive score, suggesting that Pub2Priv can effectively leverage relevant public knowledge. Interestingly, TSTR discriminative scores are similar for different correlations, potentially due to the diversity in random private samples, which makes classification inherently harder.

### G.2  Visualization of synthetic data

Here provide a visual comparison between original data and synthetic data generated by our model in fig. 8. The time series generate by our model closely align with the original holding positions.

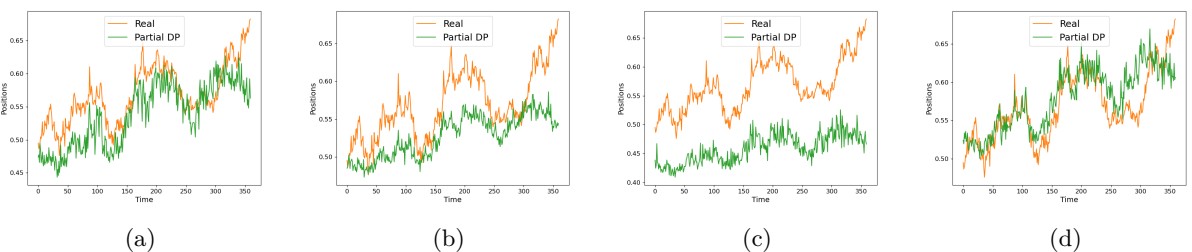

|  (a)  |  (b)  |  (c)  |  (d)  |

Figure 8: Portfolio time series generated by Pub2Priv given the same metadata condition $c$.

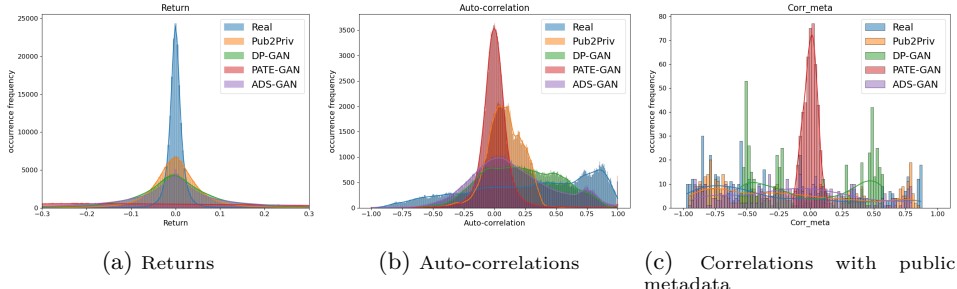

(a) Returns  (b) Auto-correlations  (c) Correlations with public metadata

Figure 9: Distributions of return, auto-correlation, and the correlations with metadata.

| Input time series length | 100 | 200 | 300 |
|---|---|---|---|
| Pub2Priv | 2m16s | 2m22s | 3m21s |
| DP-GAN | 3m32s | 4m16s | 5m29s |
| PATE-GAN | 2m04s | 2m04s | 3m30s |

Table 6: The runtime of Pub2Priv, DP-GAN, and PATE-GAN for input time series with different length.

| Data size (dimension × length) | 1×100 | 1×200 | 1×300 | 2×100 | 2×100 | 3×100 |
|---|---|---|---|---|---|---|
| Peak GPU memory usage (MB) | 1208 | 1564 | 5764 | 1422 | 1970 | 2748 |

Table 7: The computational cost of Pub2Priv for input time series with different size.

### G.3    Additional Time series utility metrics

In addition to the KS statistics, here we take a closer look at fig. 9 which shows the distribution of time series stylized facts including return, auto-correlation, and private-public data correlations. We observe that Pub2Priv yields synthetic data with stylized facts distributionally close to the original dataset.

### G.4    Scalability and Computational Cost Analysis

All experiments in the paper were conducted on AWS g4dn.4xlarge instances (16 vCPUs, 64 GB RAM, 16 GB GPU). Here we further investigate the computational cost (GPU usage) and the runtime of our models w.r.t. the size of the input data (dimension and length of ). Each dataset contains 500 samples, and we train all models with batch size 32 for 100 epochs.

We show the results averaged over five runs in table 6 and table 7. The GPU usage of all models are very similar as we match the size of the generator networks, and the runtime of our model scales comparably for varying data dimensionality.

