# OpenReview forum: "Privacy-Aware Time Series Synthesis via Public Knowledge Distillation"
_TMLR — Accepted by TMLR_

### Review · Reviewer_X6Dd · 2025-07-29

**Summary Of Contributions:**

In this work the authors propose a new way to generate private synthetic data. The proposed method requires using public context data, which is the same size as the private data, as context for conditional diffusion training with the private data. A text embedder needs to be trained using the public dataset to get the context for diffusion. Empirical comparison with other prior methods suggest that the proposed method achieves stronger privacy utility tradeoff. The author further propose an empirical DP metric called identifiability.

**Audience:**

Yes

**Claims And Evidence:**

Yes

**Requested Changes:**

- Explanation to the weaknesses part, ideally add discussion in the paper.
- Does the author try different clipping bound for the same epsilon? If not how does the author ensure that the model is pareto optimal in terms of privacy utility tradeoff.
- Provide a practical use case for identifiability would be helpful.

**Strengths And Weaknesses:**

Strengths:
- In general I think the topic is interesting and the method seems sound to me.
- Empirical performance seems quite strong.

Weaknesses:
- The method requires lots of public data. This has two issues: 1. it is unclear how similar the public data is to the private data. If there are a lot of similarity between the two, it would not be surprised that the utility is good but meanwhile it's hard to argue how much privacy about the private is leaked during public training of the embedder as the two distribution could overlap. 2. Correct me if wrong, training of GAN does not seem to require the public data, which makes the comparison with GAN unfair (as we are using more data here).
- I'm a bit concerned about the empirical privacy metric of identifiability. Although the authors claim that identifiability is positively correlated with epsilon, it cannot be viewed as a proxy / a predictor / a way of auditing for DP. For example, identifiability=0.25 could mean eps=0.01 for AIM, and eps>>10 for Pub2Priv. Hence, I'm a bit concerned about what the practical usage of the identifiability metric is.

---

> ### Author Response · Authors · 2025-08-24
> **Response to Reviewer X6Dd**
>
> Thank you for your comments. Below are our responses:
> - **Concerns about public data.** Unlike prior semi-private works (e.g., Alon et al., Lowy et al., Wang & Zhou) that use additional public data as an in-distribution or out-of-distribution dataset, we consider public information as *heterogeneous knowledge*. For example, stock markets and weather conditions are used to facilitate the generation of sensitive investment activities or household electricity consumption. These public signals do not contain or leak any private records, and there is no distributional overlap with the private time series since they are fundamentally different variables. However, similar to all semi-private approaches, our framework benefits more from public knowledge that is informative. To evaluate robustness, we explicitly control the public–private interconnection and report results in Sec. 6.5 and App. G.1, and also discuss limitations in Sec. 7.
> - **Concerns about GAN baselines.** As we described in Appendix E, we concatenate public metadata as conditional input to adapt the baselines models for our problem settings. We did not adapt baselines with our encoder $\theta_T$ or the partial DP-SGD procedure, since these constitute the core contributions of our method.
> - **Use case of identifiability.** DP provides strong theoretical guarantees but may still result in synthetic data that appears identifiable to real individuals in practice. This is because DP offers strong privacy guarantees by ensuring that the inclusion or exclusion of a single data point does not significantly affect the model's output. In a very extreme scenario, if a dataset consists only of *duplicates of one individual Bob*, the model may learn to generate only Bob-like samples, which is still DP (removing one Bob from the training data does not change the trained model) but reveals sensitive information. Our identifiability score is designed to capture this *empirical* notion of privacy leakage by checking whether synthetic samples are too close to real samples, complementing formal DP guarantees. Thus, DP controls privacy at the mechanism level, while identifiability helps interpret *practical risk* in generated samples. We clarified this distinction and added discussion in Sec. 6.4.
> - **Clipping bound.** We explored $C \in \lbrace 0.1, 0.5, 1.0, 1.5, 2.0 \rbrace$ and selected the value with the lowest validation loss to balance gradient clipping distortion and added noise. While this is not an extensive search, DP-SGD engines such as Opacus are generally resilient to clipping, and the model quickly adapts so that its gradients stay below the clipping threshold after the first few iteration, which is why some prior works (e.g., Dockhorn et al., 2022) simply fix $C=1$. We have clarified our settings in the paper.

---

### Review · Reviewer_Ygz9 · 2025-08-02

**Summary Of Contributions:**

This paper proposes incorporating public metadata for pirvacy-aware time-series data generation. Self-attention is used encode public data into temporal and feature embeddings. These are fed into diffusion models as conditional inputs and used for data generation.

**Audience:**

Yes

**Claims And Evidence:**

Yes

**Requested Changes:**

see above for questions and comments

**Strengths And Weaknesses:**

### Strengths
- The paper solves an interesting and relevant problem, is quite well-written and easy to understand.
- The experiment section is quite extensive.

### Weaknesses

### Section 3
- In (5), what is $X$? The distribution of actual data samples? If yes, please clarify it.

### Section 4
- $D_{x,c}$ has $n$ i.i.d. data samples. But, how can time series data be i.i.d.? If it is i.i.d., it's not an interesting time series, and is no longer a challenging problem, right? Or are you considering multiple instances of the entire time series ($L \times F$) as i.i.d.?

### Section 5
- It is said on page 5 that "the novelty of our work lies in problem formulation itself ..." Is the problem formulation in the paper first of its kind?
- Section 5.3: "$\theta_T$ is a pre-trained model that have no access the public metadata data ..." Do you mean "private" data?
- In Algorithm 1, why is noise added (step 9) after averaging the individual gradients? Averaging reduces the variance of the stochastic gradients. If we added noise to individual stoch. gradients, we would perhaps need to add less noise. What am I missing? Also, what is $B,n$?

### Section 6
- This is perhaps a basic question: are measures of distributional similarity like t-SNE related to more classical distribution distances like TV, KL divergence or Wasserstein distances?
- Section 6.4: "proportion of synthetic data that might be “identifiable" to the real data, where less identifiability indicates stronger privacy." I don't understand this: on one hand, we want the synthetic data to be close in terms of distribution to the real data. On the other hand, we want them to be farther in distance? Are these not slightly contradictory objectives? And the choice of Euclidean distance as the objective to measure identifiability also seems arbitrary. Why not distributional distance?
- Section 6.5: "However, while the baseline exhibits an exponential rise in TSTR loss, ..." Where do we see this exponential rise? Is this a precise mathematical statement?

---

> ### Author Response · Authors · 2025-08-24
> **Response to Reviewer Ygz9**
>
> Thank you for the detailed and constructive review. We respond to the comments and questions as follows:
>
> Section 3.
> - Thanks for the question. In Eq. (5), $X$ denotes the empirical data distribution of clean sequences. We have revised Sec. 3 and rewrite Eq. (5) to clarify this.
>
> Section 4.
> - The $n$ i.i.d. samples refer to entity-level pairs $(x_i,c_i)$, where each $x_i\in\mathbb{R}^{L\times F}$ is a full multivariate time series (e.g., investments on 4 stocks over 365 days); the i.i.d. assumption holds across entities only (e.g, $n$ investment accounts in the dataset are i.i.d). The time points and channels within $x_i$ are not assumed i.i.d. We have revised Section 4 to clarify this and avoid confusion.
>
> Section 5.
> - Yes. Prior semi-private learning (e.g., Alon et al., 2019; Lowy et al., 2024) treats public data as an auxiliary dataset (e.g., assuming some clients/patients are not sensitive and publicly available). We consider a new angle of using heterogeneous public knowledge (e,g., economics/environments) to condition a DP generator, while enforcing DP only on the private sequences. To our knowledge, this problem setup is first-of-its-kind.
> - Thank you for the careful reading. Yes, we have fixed it in the paper.
> - In our paper, $n$ is the size of the training dataset and $B$ is the size of the mini-batch (a.k.a. Lots in Abadi et al. 2016). Mini-batches are sampled at each step of DP-SGD for calculating the bounds for gradient clipping and noise addition.
> DP noise is calibrated to the $\ell_2$ sensitivity of the released aggregate after clipping. Thus, following the Gaussian mechanism, DP-SGD adds $\mathcal N(0,\sigma^2 C^2 I)$ to the sum (Abadi et al., 2016; Xie et al., 2018; Dockhorn et al., 2022). Dividing by $B$ then yields the noisy average, which is equivalent to adding $\mathcal N(0,\tfrac{\sigma^2 C^2}{B^2} I)$ directly to the average. We noticed a missing parenthesis in the line (9) and have fixed the issue and clarified $B$ and $n$ in the paper.
> If instead noise were added to each clipped per-example gradient and then the sum, $\hat g = \frac{1}{B}\sum_{i\in B}(\bar g_i + \xi_i), \quad \xi_i \stackrel{\text{i.i.d.}}{\sim} \mathcal N(0,\tau^2 I),$
> the noise on $\hat g$ would have covariance $\tfrac{\tau^2}{B} I$. To match the calibrated mechanism on the average ($\mathcal N(0,\tfrac{\sigma^2 C^2}{B^2} I)$), one must set $\tau^2 = \tfrac{\sigma^2 C^2}{B}$. This is algebraically equivalent to adding noise once to the sum, and does not reduce the total noise required for a given privacy target.
>
> Section 6.
> - t-SNE is not a formal measure of distributional similarity in the sense of classical distances such as TV, KL divergence, or Wasserstein distance. However, it has been widely used in time series literature as a non-linear dimensionality reduction technique to visualize multivariate synthetic time series. We have clarified in the paper that t-SNE only serve as an illustrative tool to give intuition about coverage, while our quantitative evaluation relies on formal metrics such as $KS_R$, $\mathrm{KS}_{AR}$, and TSTR.
>
>     [1] Coletta et al. “On the Constrained Time-Series Generation Problem”, NeurIPS 2023.
>
>     [2] Yuan et al. “Diffusion-TS: Interpretable Diffusion for General Time Series Generation” ICLR 2024.
>
>     [3] Naiman et al. “Utilizing Image Transforms and Diffusion Models for Generative Modeling of Short and Long Time Series”, NeurIPS 2023.
> - This is a good question. While we want the synthetic dataset $D' = \lbrace x'_1, \dots, x'_n \rbrace$ to be close to the real dataset $D = \lbrace x_1, \dots, x_n \rbrace$ in terms of overall distribution, each synthetic datapoint has a risk of exposing information of a real individual (i.e., $\exists i, j$ such that $x'_i \approx x_j$). To capture this notion, we define the identifiability score $I(D,D')$ using **pairwise** distances: lower identifiability means **fewer synthetic points are nearly identical to real samples**, while the distributions can remain aligned. In this sense, distributional similarity and pointwise identifiability are complementary rather than contradictory. We acknowledge that here the choice of pointwise distance is rather practical, and other distance metrics could be explored in future work for assessing empirical privacy for DP generators.
> - We apologize for the unrigorous description. We have revised it to “the baselines exhibit a much steeper rise in TSTR loss,” which more accurately reflects the trend illustrated in Fig. 6.

---

### Review · Reviewer_tt2c · 2025-08-11

**Summary Of Contributions:**

The authors Pub2Priv, a framework for generating private time series using public contextual metadata through conditional diffusion models trained with DP-SGD.

**Audience:**

No

**Claims And Evidence:**

No

**Requested Changes:**

I am afraid that applying standard conditional diffusion models with DP-SGD using auxiliary data does not meet the bar for TMLR. I would encourage the authors to think about the limitations of their work (some of which are mentioned in section 7, and section 6.5) from a fresh perspective.

**Strengths And Weaknesses:**

The problem of generating private time series using public metadata is a useful problem. But I think there are several weaknesses with the state of the draft as is,

1. The authors mention that "no prior work has explored the use of public contextual information for time series generation with differential privacy." This is a rather strong claim for a problem statement that seems quite natural to study. Indeed a quick search shows quite some prior work (see below)

* Differentially Private Generative Adversarial Networks for Time Series, Continuous, and Discrete Open Data, Frigerio et al.
* Achieving Privacy Utility Balance for Multivariate Time Series Data, Hore et al.
* Differentially private multivariate time series forecasting of aggregated human mobility with deep learning: Input or gradient perturbation?, Arcolezi et al.
* Optimal Differentially Private Model Training with Public Data, Lowy et al.
* Privacy Amplification by Structured Subsampling for Deep Differentially Private Time Series Forecasting, Schuchardt et al.


2. The authors spend a lot more time explaining Diffusion models and DP-SGD than I would have liked -- it comes at the cost of several missing details about the stuff that actually matters. For instance,
* What is the CSDI architecture?
* Are positional embeddings used?
* What is the loss for $\theta_T$?
* The authors mention "appendix" several times. Which appendix? Please label them for the reader
* What is k in 6.3.1, is it 4?
* Please elaborate/explain/discuss the key metrics KSR, KSAR, TSTR etc

3. One of the evaluation metrics |Corr_meta| directly measures what the model is designed to optimize. So not so surprising that the proposed model does well than the baseline.

4. Theorem 5.1 adds little value to the reader.

5. A figure is worth a thousand words. It is always a good idea to show tables/plots/figures then verbally stating results c.f section 6.3.2

6. Section 6.5 -- "leading to a decline in performance for all models". A rather important claim that is not backed by numbers for all 3 datasets (and only for small-ish k for investment dataset). If I am mistaken please point me to the right page.

7. In my humble opinion, t-sne plots for contrasting different generative models especially when the original dimensions are so high, should be used with extreme caution since they could be so easily misinterpreted.

8. Could the authors please comment on why they used diffusion models especially when autoregressive models have been used quite extensively recently for time series modeling? [https://neurips.cc/virtual/2024/workshop/84712]

---

> ### Author Response · Authors · 2025-08-24
> **Response to Reviewer tt2c**
>
> Thank you for your feedback and we address the specific comments below.
> 1. Thanks for pointing out these prior works, They are related works and here is a breakdown of the papers that you mentioned:
> | Paper| Time-series generation?| Utilizes public knowledge?| Differential privacy?|
> |-|-|-|-|
> | Frigerio et al. (2019)| Yes| No| Yes|
> | Hore et al. (2024)|No—data release|No|No (LIP)|
> | Arcolezi et al. (2022)|No—forecasting|No|Yes|
> | Lowy et al. (2024)|No|No—public data in the same type|Yes|
> | Schuchardt et al. (2025) |No — forecasting| No| Yes|
> | **Ours**| **Yes**| **Yes**| **Yes**|
>
> Lowy et al. (2024) is the only work that leverages public information, which we already discuss in the Related Work section. As noted in Sec. 1 (para. 2) and Sec. 2.2 (para. 2), prior semi-private learning only considered public data as an auxiliary dataset rather than heterogeneous, per-sequence contextual conditions for generation. Consequently, the specific problem we study remains unaddressed. We have added the remaining papers to the Related Work.
>
> 2. We clarified missing details in the revision:
> - CSDI is a dual-axis transformer encoder (Tashiro et al., 2021). We expanded Sec. 5.1 to explain it.
> - Yes, we added positional encoding. Details are in the updated Sec. 5.1.
> - We pretrain $\theta_T$ via masked reconstruction. full loss is in Sec. 5.1.
> - We revised the text to properly link to the corresponding appendix.
> - We didn’t find $k$ in Sec. 6.3.1 but we believe you’re referring to Sec. 4 and Sec. 5.1 where $k$ denotes the number of public channels. It is 7, 2, and 1 for the portfolio, electricity, and comtrade dataset correspondingly (App. C).
> - All metrics (KS$R$, KS${AR}$, TSTR) are defined in Sec. 6.3.1, and we added their full equations in Appendix F.
> 3. We introduce $|Corr_{meta}|$ purely as a diagnostic of whether the conditional dependence between $x$ and $c$ is preserved; it does not appear in the losses for $\theta_T$ or the diffusion model $\theta_{DM}$. Beyond $|Corr_{meta}|$, our method shows consistent gains on distributional metrics ($\mathrm{KS}R$, $\mathrm{KS}{AR}$) and privacy-utility trade-off based on TSTR.
> 4. Agree. We removed it from the main text.
> 5. Utility results are supported by Table 1, Fig. 3, and Fig. 4. We revised Sec. 6.3.2 to improve readability.
> 6. Section 6.5 studies how performance changes as the private data to be generated becomes more complex while the same public metadata given remains the same. We control the diversity and complexity of the private portfolios by increasing the number of investment strategies. Our intent was not to claim a universal “decline for all models” across all datasets, but rather to demonstrate robustness: even as the number of strategies grows, our method remains the top performer (Fig. 6). While this manipulation applies only to the portfolio dataset, we believe it is sufficient to demonstrate our model’s performance, as it is the largest dataset and has the largest number of public channels (k=7). We notice the confusion and have revised Sec. 6.5 for clarification.
> 7. While t-SNE plots are commonly used in many SOTA models (GuidedDiffTime, ImagenTime, Diffusion-TS) for time series generations (considering it is hard to visualize multivariate time series), we agree with your careful concerns on its interpretation. Therefore we have revised Sec. 6.3.2 and highlight that we use t-SNE **only as a qualitative visualization** to give intuition about coverage, in addition to our quantitative metrics.
> 8. We chose diffusion as recent diffusion variants have demonstrated strong, often state-of-the-art performance on unconditional/conditional generation, constrained generation, and multi-resolution modeling for sequences, e.g., TSDiff (NeurIPS 2023), TimeDiff (ICML 2023), Diffusion-TS (ICLR 2024), Multi-Resolution Diffusion (ICLR 2024), and Time Weaver (ICML 2025). Most importantly, although this paper focuses on time-series generation, adopting a general and flexible diffusion-based framework makes Pub2Priv beneficial for the ML research community on addressing sensitive data in other modalities (e.g., images, tabular).
>
> Finally, we appreciate the feedback respectfully disagree with the characterization of our work. Specifically: (1) we consider **a new angle of using public knowledge** (economics/weather condition) rather than a public dataset in the same type (investment/electricicy usage); (2) while DP-SGD and diffusion model are known, we develop a unique framework that separates a public-knowledge encoder from DP-SGD so **noise is applied only w.r.t. private data**. While we acknowledge that our work has several limitations as discussed in Section 7, we believe that according to the TMLR acceptance acceptance criteria, https://jmlr.org/tmlr/acceptance-criteria.html, our novel problem formulation and unique framework show direct relevance to the audience interested in privacy-preserving generative modeling, and the experiment results provide sufficient evidence.

---

### Author Response · Authors · 2025-08-24
**Thank you for your feedback**

We thank all reviewers for their constructive and helpful feedback. We uploaded a revised version of the paper and highlighted changes in blue  for convenience.

---

### Decision · Action_Editor_xAQs · 2025-10-19

**Recommendation:** Accept with minor revision

**Additional Comments:**

Reviewers all acknowledged that the submission tackles an interesting and important problem, but there were concerns about the level of technical innovation in the work, as well as issues with presentation. The revision has resolved many presentation-related issues though some still remain: these are listed below and should be addressed for the camera ready version of the paper.

Overall, in spite of the lack of significant technical novelty, as the paper's claims are supported by evidence and the work is likely to be appreciated by at least some in the community, it meets the bar for TMLR.

Required revisions:
- Fix the references to use parenthetical references unless authors' names are used in the sentence
- Section 5.2 "computes the gradients
(where B)" - the definition of B is missing and should be added
- Section 6.2.1 - fix circular reference to section 6.2 "(more information of the dataset can be found in section 6.2)"

**Audience:**

Yes

**Audience Explanation:**

This paper considers the problem of generating synthetic data that is useful yet privacy-preserving. The methods proposed in this work should be of interest to those in the community who work on sensitive applications in domains like healthcare and finance.

**Claims And Evidence:**

Yes

**Claims Explanation:**

The primary claim of this submission is a method that leverages publicly available non-sensitive data to improve the generation of synthetic time-series data, with the goal of improving the utility for downstream analyses while preserving privacy. This claim is validated through experiments on three different time-series datasets from varied domains, as well as in ablation studies that show the utility of the additional public data and design choices. It also claims to propose a metric for assessing identifiability of the synthetic data, which is validated in the experiments as well.